# Security and Privacy Issues in IoT-Based Big Data Cloud Systems in a Digital Twin Scenario

**Christos L. Stergiou** *[ID], **Elisavet Bompoli** and **Konstantinos E. Psannis** *[ID]

Department of Applied Informatics, University of Macedonia, 54636 Thessaloniki, Greece
* Correspondence: c.stergiou@uom.edu.gr (C.L.S.); kpsannis@uom.edu.gr (K.E.P.); Tel.: +30-2310-891-737 (K.E.P.)

**Abstract:** Due to its unique type of services, Cloud Computing could operate as a "base technology" for other technologies; this attracts researchers to develop sustainable Cloud systems. It is a new generation of services that offers an opportunity for users to access and manage their information, applications, and data regardless of place and time. Nevertheless, there is a type of service that can include large amounts of data, called Big Data, and it consists of the rapid use of the Internet of Things (IoT) to produce large data sets. In this work, initially, we present Cloud Computing (CC) and Big Data (BD) exported from IoT, focusing on the security and management challenges of both. Notably, we combine the two aforementioned technologies to examine their related characteristics and discover new perspectives and opportunities for their integration and to achieve a sustainable environment called a Digital Twin scenario. Subsequently, we present how Cloud Computing contributes to IoT-based Big Data, aiming to fill a scientific gap in the sector of their integration regarding security and privacy. Finally, we additionally survey the security challenges of the integrated model of BD and CC and then propose a novel security algorithm for sustainable Cloud systems in a Digital Twin scenario. The experimental results presented are based on the use of the encryption algorithms AES, RC5, and RSA, and our proposed model extends the advances of CC and IoT-based BD, offering a highly novel and scalable service platform to achieve better privacy and security services.

**Keywords:** Big Data; Cloud Computing; Internet of Things; security; privacy; algorithm; integration; Digital Twin





## 1. Introduction

As we already know, Cloud Computing can offer several important features to users [1]. In particular, Cloud providers and customers are keen to build a more secure Cloud environment due to its unique function. In more detail, the following are the delivery data models of Cloud Computing [1,2]: *Software as a Service (SaaS)*, *Platform as a Service (PaaS)*, and *Infrastructure as a Service (IaaS)*. Additionally, the major characteristics of Cloud Computing (*Cloud Computer Features—CCF*) are Storage over Internet (*CCF1*), Service over Internet (*CCF2*), Applications over Internet (*CCF3*), Energy Efficiency (*CCF4*), and Computationally Capable (*CCF5*).

Nowadays, there are tremendous amounts of data generated daily in the sectors of manufacturing, business, science, and people's personal lives. These data, if processed properly, could reveal new knowledge about the market, society, and environment, additionally enabling people to cope with emerging opportunities and changes promptly [3,4].

Furthermore, the accustomed data processing technologies, such as database sets and data warehouses, are becoming inadequate for the huge amounts of data that need to be dealt with. These challenges are known as Big Data (BD), and they constitute a novel field of study for researchers [3,4].

Big Data is characterized as "*a big thing in the field of modern technologies*" [4]. The five Vs of BD are **Volume**, **Velocity**, **Variety**, **Veracity**, and **Value** (Figure 1). All the data related to the term Big Data have a specific origin or "source", which, based on Variety, could give

various data types. Here, we attempt to address several major Big Data sources (*BDSs*), as well as the respective challenges that arise from them regarding the overall use of Big Data.

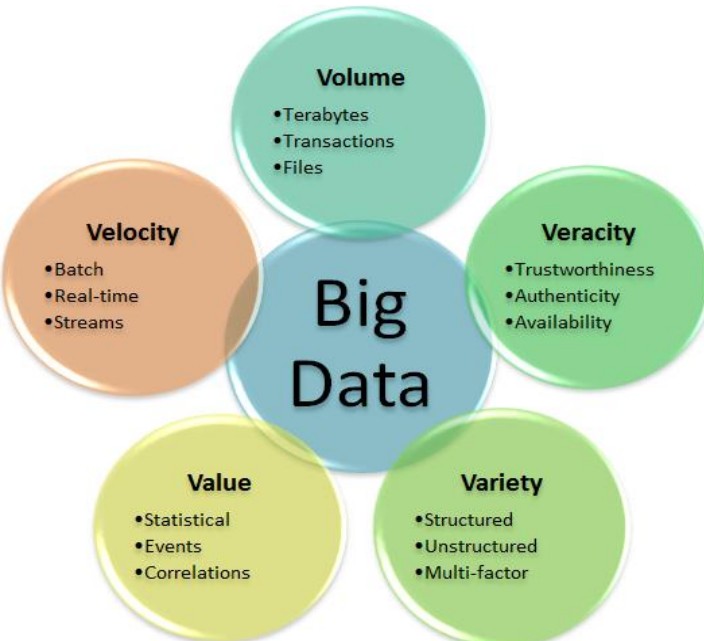

**Figure 1.** The 5Vs of Big Data and their major characteristics.

➤ *BDS1—Earth, Marine, and Space Sciences:* Large data sets are collected and generated every second and at different space–time scales for operations, as the presentation, monitoring, and understanding of complex earth, marine, and space systems are enabled by the preference of sensing and computing simulation technologies. So, as an example, earth, marine, and space observation software collects terabytes of images daily [5–8], with gradual increases in space, time, and spectral analyses [5,9].

➤ *BDS2—Internet of Things:* The IoT is a broader aspect and is considered to be all the devices that would be able to connect to the internet and could interact with each other [1,10–12]. The whole data that can be generated from the various IoT sensors includes spatiotemporal information, and thus, it can be described as BD. The combined use of IoT-BD in network environments, in addition to being integrated with technologies such as Cloud Computing, could offer new opportunities and lead to the accelerated development of Smart Cities [5,13].

➤ *BDS3—Social Sciences:* Big Data are being generated by various social networks, such as Instagram, Twitter, and Facebook, and thus they could transform social sciences [5].

➤ *BDS4—Business:* The various decisions for strategy, managing optimization, and competition related to Big Data could be enhanced by business intelligence and analytics [5]. Data related to the previous scenarios contain harmful amounts of geospatial information, for example, where and when a transition occurred [14,15].

➤ *BDS5—Industry:* In Industry 4.0, which is considered the fourth industrial revolution, the products and production systems leverage technologies such as IoE-based BD, aiming to establish ad-hoc networks for self-managed applications [16,17].

To operate and verify our proposed scenario and to evaluate our proposed algorithm, we used a Digital Twin scenario with the goal of simulating the operation of a Cloud server, through which it manages, transfers, and produces IoT-based Big Data. We selected the option of Digital Twin because it offers more reliable results for our proposal through its "simulated" scenario. We use the Digital Twin concept in our proposal to better predict the functionality of our proposal through a virtual model designed to accurately reflect a physical object. After that, we could extract useful information about the reliability of our

proposal and the degree of improvement of the existing situation. Our evaluation scenario was tested on CloudSim software, using all the necessary features.

To sum up, the significant contributions of our research include:

✓ Presenting Cloud Computing and IoT-based Big Data that focus on security and management challenges in a more sustainable environment, in a Digital Twin scenario.
✓ Integration benefits of Cloud Computing and IoT-based Big Data in a Digital Twin procedure.
✓ The use of the encryption algorithms AES, RC5, and RSA and the proposed model extend the advances of Cloud Computing and IoT-based Big Data, offering a highly novel and scalable service platform to achieve more secure services in the sustainable environment of a Digital Twin scenario.
✓ The filling of a scientific gap in the sector of integrating CC and IoT-based Big Data.
✓ Proposal of a novel security model and an algorithm for sustainable Cloud systems that offer a more secure use of BD in CC as an integrated model of these two technologies.

The rest of the paper is divided as follows. Section 2 presents the background studies in the field of CC and IoE-based BD integration. Section 3 provides an analysis of the background studies, aiming to address the "*gaps*" and the current challenges of Cloud Computing and IoT-based Big Data integration. Section 3 also lists in a table the significant challenges that were distinguished from the related studies. Section 4 offers a brief analysis of Cloud Computing's and Big Data's challenges, along with the main security issues. Section 5 offers the model and scenarios of the integration of Cloud Computing and IoT-based Big Data. Next, Section 6 compares our proposed method with other proposed methods in this field. The simulation results are based on a practical system, and an experimental analysis is demonstrated in Section 7. Subsequently, Section 8 summarizes the whole study, along with some directions for future research.

## 2. Background Research

A large number of research works have been conducted in recent years to integrate Cloud Computing with Big Data. So, for this study, we studied and analyzed previous research in the literature that investigated the integration of Cloud Computing technology with Big Data technology [2,4,5,18–24]. All the works are presented here in ascending chronological order.

First, Takabi et al. [2], in their work, explored roadblocks and solutions, aiming to provide a trustworthy CC environment.

Agrawal et al. [18] introduced a type of tutorial work, which is an organized picture of the issues faced by application developers and DataBase Management Systems (DBMS) designers in developing and deploying internet-scale applications.

In their work, Demirkan and Delen [19] proposed a conceptual framework for DSS in Cloud and discussed research directions, taking into account a list of requirements for service-oriented DSS, which they defined.

Castelino et al. [20] presented stresses on the integration of BD with CC, which can serve as a driving force for the business and IT industry and for data analytics in general.

In their work, Inukollu et al. [21] introduced a discussion about security issues for CC, BD, Map-Reduce, and Hadoop environments. In particular, the main focus of the work of Inukollu et al. was on the security issues of CC that are associated with BD.

Hashem et al. [22] reviewed the rise of BD in CC and discussed the relationship between them, BD storage systems, and Hadoop technology.

Yang et al. [5] studied BD and CC and reviewed the advances and the consequences of utilizing CC to tackle BD in the digital earth and relevant science domains.

Stergiou and Psannis [4] studied BD and CC along with their features, focusing on the security and privacy challenges of both of them and combining their functionality, aiming to examine the common characteristics and discover benefits concerning security issues.

Furthermore, Stergiou and Psannis [23], in another work, studied BD and CC technologies, in addition to their basic features, focusing on privacy and security challenges

and trying to determine another aspect of combining the functionality of two technologies, with the aim of examining the benefits related to security challenges of their integration. Summarizing their work, the authors presented a new algorithm that can improve CC's security using algorithms that provide security in BD.

Pargmann et al. [24] introduced an approach in which a sound semantical integration of several information types was shown and applied to a concrete use-case scenario.

Aguzzi et al. [8], in their research, focused on marine biomimetic research and used innovative bibliographic statistics tools in order to highlight established and emerging knowledge domains. In the research environment of their work, they identified natural processes by which living organisms obtain energy and are thus urgent to sustain energy-demanding tasks, while, at the same time, the natural designs must increasingly inform the optimization of energy consumption.

## 3. Background Research Analysis

Taking into account Section 2, we determined that the study of finding and achieving an integrated model/method of Cloud Computing technology with IoT-based Big Data technology has become more popular in the academic and research community in recent years.

Table 1 presents the challenges presented and, in most cases, have been addressed by the related work that we studied. Most of the works listed in Table 1 focus on the "*Management*" challenge, which, in our opinion, is one of the major issues of Cloud Computing and Big Data integration. The second most popular challenges of the related works are "*Computation (Processing) and Analysis*", "*Security*", and "*Privacy*" issues. In more detail, out of the 11 related works we studied, the statistical results are as follows: Privacy—7 of 11, Security—7 of 11, Storage—6 of 11, Access Control—7 of 11, Computation (Processing) and Analysis—7 of 11, Management—9 of 11, Reliability—6 of 11, and Scalability—3 of 11. Equally as important, and as we can observe from the statistical analysis, the less-mentioned issues are "*Scalability*" and "*Storage*", which are vital for the functionality of both technologies. Regarding the statistical analysis of our research, and taking into account the recent works that have been carried out in the sector of Cloud Computing and Big Data, we realized that more research is needed to find better solutions to improve the security and privacy issues of the integration of these technologies.

**Table 1.** Background research challenges in the field of the integrated models/methods of Cloud Computing technology and Big Data technology.

| Related Background Research | Privacy | Security | Storage | Access Control | Computation (Processing) and Analysis | Management | Reliability | Scalability |
|---|---|---|---|---|---|---|---|---|
| Takabi et al. [2] | √ | √ | | √ | | √ | √ | |
| Agrawal et al. [18] | √ | √ | √ | | √ | √ | | √ |
| Demirkan and Delen [19] | √ | | √ | √ | √ | | √ | |
| Castelino et al. [20] | | | √ | | √ | √ | | |
| Inukollu et al. [21] | | √ | | √ | √ | √ | | |
| Hashem et al. [22] | √ | √ | √ | √ | √ | √ | √ | √ |
| Yang et al. [5] | √ | √ | √ | √ | √ | √ | √ | √ |
| Stergiou and Psannis [4] | √ | √ | | | | | √ | |

**Table 1.** *Cont.*

| Related Background Research | Challenges | | | | | | | |
|---|---|---|---|---|---|---|---|---|
| | Privacy | Security | Storage | Access Control | Computation (Processing) and Analysis | Management | Reliability | Scalability |
| Stergiou and Psannis [23] | √ | √ | | | | √ | | |
| Pargmann et al. [24] | | | | √ | √ | √ | √ | |
| Aguzzi et al. [8] | | | √ | √ | | √ | | |

## 4. Cloud Computing and Big Data Challenges

### 4.1. Privacy and Security Challenges in Cloud Computing

As we determined from the literature review, Cloud Computing environments can be characterized as multi-domain environments. These environments could protect each domain as they can use different security, privacy, and trust requirements, and could lead them to employ various mechanisms, interfaces, and semantics [2]. Thus, the major challenges of Cloud Computing are associated with the concept of user authentication and identification, access to and the management of data used, and, as a result, the security and privacy management of data, with the associated policies and data protection rules. These challenges identified in the literature are (*Cloud Computing Challenges—CCC*): Authentication and Identity Management (*CCC1*), Access Control and Accounting (*CCC2*), Trust Management and Policy Integration (*CCC3*), Secure-Service Management (*CCC4*), Privacy and Data Protection (*CCC5*), and Organizational Security Management (*CCC6*). We determined that the main concept of the challenges focuses on *securing the IoT-based data management services of Cloud environments*.

### 4.2. Big Data Challenges

Regarding our research, we concluded that there are three major issues of Big Data that need to be fundamentally addressed. These are storage, management, and processing (*Big Data Challenges—BDC*).

➢ *BDC1—Big Data Storage:* The quantity of data has exploded each time a new storage medium has been invented. Additionally, data creation does not have any restrictions, and data could be generated by everything connected to the network [4].

➢ *BDC2—Big Data Management:* Big Data management could be used with a focus on customizing the consistency level [25].

Moreover, *HBase* (Figure 2) is a significant implementation of the NoSQL model in the Hadoop project, which is used as a Big Data management application. *Hbase* is a distributed column-oriented database that was built on top of the Hadoop Distributed File System (HDFS) [25].

➢ *BDC3—Big Data Processing:* Let us suppose that an exabyte of data needs to be processed in its wholeness. More simply, we can assume the data is crumbled into blocks of eight words, and as a result, an exabyte is equal to *1 Kilo* of petabytes.

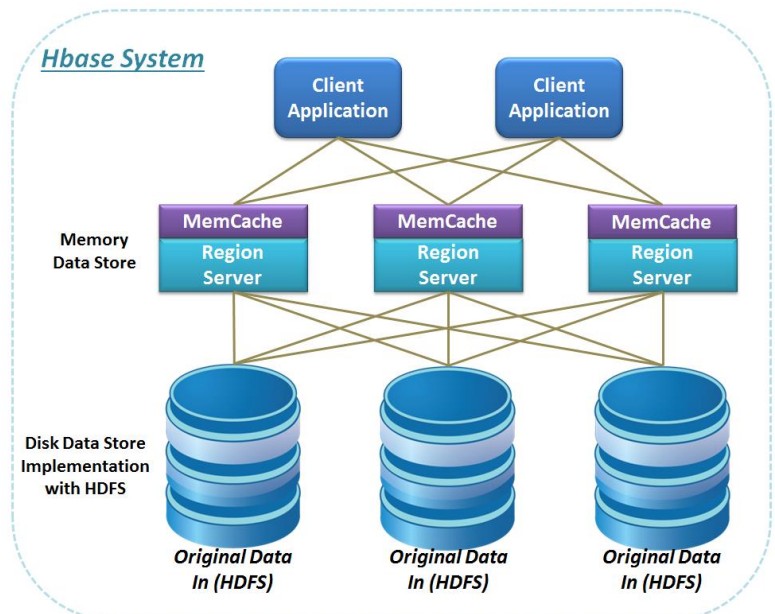

**Figure 2.** Architecture of HBase NoSQL Database System management.

## 5. Integration Aspects of IoT-Based Big Data with Sustainable Cloud Environments

To achieve the best integrated model of Cloud Computing and IoT-based Big Data, further research needs to be done due to the rapid evolution of IoT data.

Table 2 lists the basic characteristics of CC regarding the convenience this technology provides. Additionally, it enumerates the main features, also known as the 5Vs, of Big Data. The main goal of Table 2 is to demonstrate which of each feature of CC contributes more, and as a result, is related more to the features of Big Data technology. As shown in Table 2, the feature of Big Data that is influenced most by the features of CC is "*Value*". Value, as we determined previously, is related to data that consists of large data sets. Thus, it should be the most significant feature of BD contributing to the technology of CC. In regard to CC, the features that are more affected are "*Applications over Internet*" and "*Energy Efficiency*". Both of these features are based on the use of the data through the network. Applications could extract large quantities of data sets, and, additionally, the grouping of data in large data sets can lead to better use of the power resources.

**Table 2.** Contributions of Big Data to Cloud Computing.

| Cloud Computing Features<br><br>Big Data Features (5 Vs) | CCF1 | CCF2 | CCF3 | CCF4 | CCF5 |
|:---:|:---:|:---:|:---:|:---:|:---:|
| Volume | √ | | √ | √ | |
| Velocity | | √ | √ | √ | √ |
| Variety | | √ | | √ | |
| Veracity | √ | | √ | | √ |
| Value | √ | √ | √ | √ | √ |

Table 3 lists the three models of Cloud Computing technology and the basic sources of Big Data. In Table 3, we can see the necessity of using the different models of Cloud Computing for the various sources that export Big Data. Table 3 shows that the source of Big Data that contributed most to the models of Cloud Computing was "*IoT*". IoT could be considered the main source of Big Data. Additionally, its use, which is especially connected to the internet, makes it quite close to all the models of Cloud Computing technology. In regard to Cloud Computing, the models that contributed the most are "*SaaS*" and "*IaaS*".

So, we can conclude that Cloud Computing can contribute to Big Data by providing both software and hardware resources to produce large data sets.

**Table 3.** Contributions of Cloud Computing models to Big Data sources.

| Cloud Computing Models | SaaS | PaaS | IaaS |
|:---:|:---:|:---:|:---:|
| **Big Data Sources (BDS)** | | | |
| *BDS1* | √ | | √ |
| *BDS2* | √ | √ | √ |
| *BDS3* | √ | √ | |
| *BDS4* | √ | | √ |
| *BDS5* | | √ | √ |

Through the integration of BD and CC (Figure 3), we have the opportunity to extend the use, management, and transmission of large data sets that constitute the IoE-based Big Data that are provided in environments installed in Cloud environments.

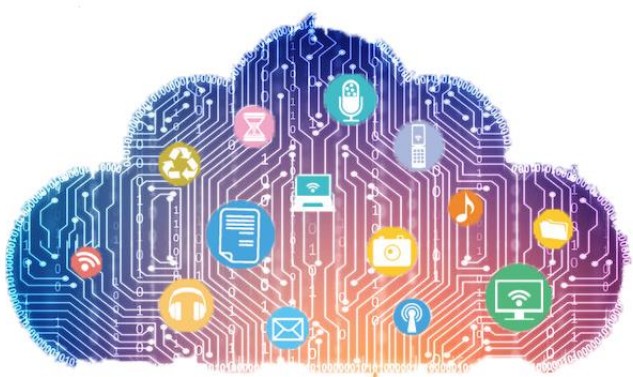

**Figure 3.** Big Data and Cloud Computing integration.

### 5.1. Challenges and Issues in IoT-based Big Data and Cloud Computing Integration

The virtual resources and unlimited capabilities of Cloud Computing that aim to balance its technological constraints, such as storage, communication, and processing, could offer beneficial uses to Big Data. Furthermore, Big Data could offer a dynamic method of delivering novel services to the world by extracting the meaning of the large-scale data sets that comprise them, taking advantage of the benefits offered by CC. In many cases, CC could offer an intermediate layer between the data and the applications, hiding all the functionalities and complexity that would be necessary to implement.

Table 4 shows the Cloud Computing and IoT-based Big Data integration issues that have been presented and, in most cases, addressed by related works that we have studied. This table shows the contributions of Cloud Computing and IoT-based Big Data integration challenges to its technology separately. More specifically, Table 4 reveals that the common challenges that affect both technologies are "*Security*", "*Storage*", and "*Management*". Taking into account the conclusion drawn from Table 1, which shows that most of the works that we have studied are focused on the "*Management*" challenge, this strengthens our view that this is the most important challenge of the integration of CC and IoT-based BD. Additionally, the other two important challenges are the storage of large data sets and making them more secure, both of which play a vital role in the integration of CC and IoT-based Big Data.

**Table 4.** Integration challenges of IoT-based Big Data and Cloud Computing.

| IoT-Based Big Data and Cloud Computing Integration Challenges | Privacy | Security | Storage | Access Control | Computation (Processing) and Analysis | Management | Reliability | Scalability |
|---|---|---|---|---|---|---|---|---|
| IoT-based Big Data | | √ | √ | | √ | √ | | √ |
| Cloud Computing | √ | √ | √ | √ | | √ | √ | |

We studied multiple works that address many significant challenges and problems regarding two procedures, which are the storage and the processing of BD in the Cloud. Nowadays, there is a small number of tools that are available to users to address the multiple challenges of BD processing in Cloud environments.

*5.2. Security Challenges of IoT-Based Big Data and Cloud Computing Integration*

There is rapid and independent development of BD and CC. Additionally, BD extends its scope to deal with various types of data by offering new techniques in various aspects of real-life scenarios, which might benefit from the use of CC. Cloud Computing could offer an intermediate layer between the users and the systems where large amounts of data exist, hiding all the complexity and functionalities [26,27].

Table 5 lists the security challenges that both Cloud Computing and Big Data face. As already analyzed in the previous section, there are three major security challenges of Big Data described in the literature (*BDC1*, *BDC2*, and *BDC3*) and six major security challenges of Cloud Computing (*CCC1*, *CCC2*, *CCC3*, *CCC4*, *CCC5*, and *CCC6*). As can be seen, Organizational Security Management is related to all three Big Data Security challenges. Thus, we easily can understand that the words "*Security*" and "*Management*" play an important role in Cloud Computing and Big Data both individually and in their integrated form.

**Table 5.** Big Data and Cloud Computing security challenges.

| **Big Data** **Cloud Computing** | BDC1 | BDC2 | BDC3 |
|---|---|---|---|
| CCC1 | | √ | √ |
| CCC2 | √ | | √ |
| CCC3 | | √ | |
| CCC4 | √ | √ | |
| CCC5 | √ | | |
| CCC6 | √ | √ | √ |

*5.3. Proposed Security Method for Big Data Encryption in Sustainable Cloud Environment in Digital Twin Scenario*

Based on the literature search, we determined that AES, RC5, and RSA are the fastest and more efficient encryption algorithms. Moreover, AES was also considered to provide a more secure environment for sustainable Cloud Computing technology in a Digital Twin scenario, based on previous studies in this field. Furthermore, the symmetric encryption method of RC5 and the asymmetric encryption method of RSA allow users to achieve a more secure Cloud environment, as tested in a Digital Twin scenario.

As we have mentioned before, the key feature of the algorithms (AES, RC5, RSA) that is important in our study is their speed. All three algorithms can be characterized as fast

algorithms due to their quick response in their encryption procedure. In contrast, one key characteristic in which they differ widely is the number of rounds of their encryption method, where AES needs 10, 12, or 14 rounds, RC5 needs 12 rounds, and RSA needs only 1 round.

Our proposed method collects and combines all the benefits of the three algorithms to provide a better encryption scenario for BD in sustainable CC environments. Thus, with our proposed model (Algorithm 1–Table 6), we can amplify the advances of Cloud Computing and Big Data technologies by offering a highly novel and scalable efficient service platform. As a result, we can propose and introduce the following algorithm for the more secure use of Big Data in sustainable Cloud Computing systems.

---

**Algorithm 1.** Proposed Algorithm

---

**Key Production Procedure**
m1 = im/8
counter = 0
while m1
    if (m1==' ' or m1 =='space')
        break
    else
        counter = counter + 1
k = im * counter

- - - - - - - - - - - - - - - - - - - - - - - - - - - - - - - - - -

**Encryption Procedure**
input k
input im
kc = 0
wpc = 0
while k
    if (k==' ' or k =='space')
        break
    else
        kc = kc + 1
while im
    if (im ==' ' or im =='space')
        break
    else
        wpc = wpc + 1
om = 1
while kc > 0
    for i = 1, i++, i≤wpc
        om = (om*i)+im
nk = k + i
transfer routine for nk and om

---

**Table 6.** Contents of the algorithm.

| | |
|---|---|
| **m1** = key production number | **wpc** = word package counter |
| **im** = input package of data | **om** = output package of data |
| **counter** = word counter | **i** = default counter |
| **k** = key | **nk** = new encryption key |
| **kc** = keyword counter | |

We decided to improve the existing benefits of the three aforementioned algorithms by proposing a novel one that combines their major benefits as a new algorithm scenario. To prove the functionality of the proposed algorithm, we conducted several simulations with different amounts of data and over various time scales to better study the complexity

of our algorithm. The experimental scenarios and results are illustrated in Section 7, and they show how the proposed algorithm operates better than the existing algorithms. We present four different scenarios (Figures 4–7), showing four different combinations of data and time.

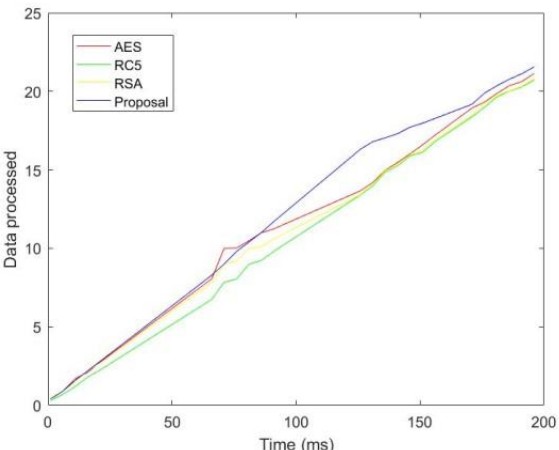

**Figure 4.** Performance of data processed (megabytes) comparison of the AES, RC5, RSA and proposed model.

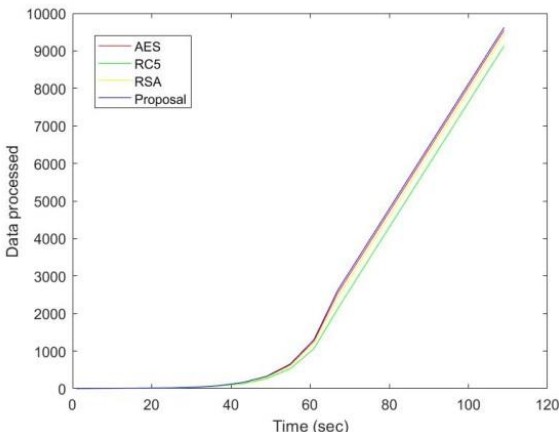

**Figure 5.** Performance of data processed (megabytes) comparison of the AES, RC5, RSA, and proposed model.

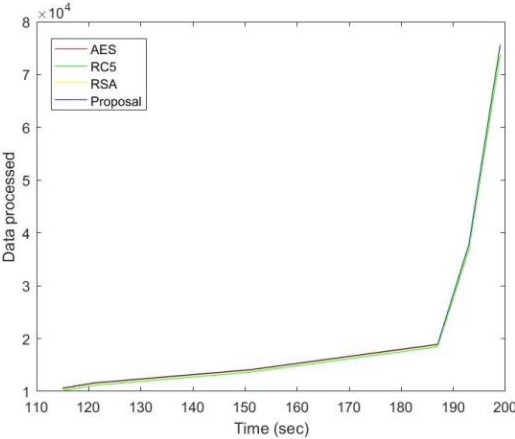

**Figure 6.** Performance of data processed (megabytes) comparison of the AES, RC5, RSA, and proposed model.

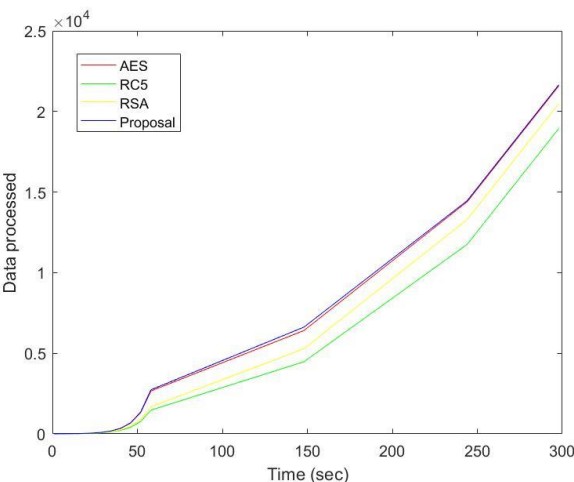

**Figure 7.** Performance of data processed (megabytes) comparison of the AES, RC5, RSA, and proposed model.

## 6. Comparative Analysis

In addition to the literature review of past works presented in Section 2, we also reviewed some other significant works in the specific field of CC and BD integration, with the aim of security. This section presents a comparative analysis of the related works that we have distinguished in the following paragraphs. Initially, we analyzed what each of them deals with, presented from the oldest to the most recent.

K. Gai et al. [28] proposed D2ES, which is a new data encryption approach, focused on selective data encryption regarding privacy classification schemes under time limitations. S. K. Mishra et al. [29] proposed an algorithm with adaptive task allocation concerns that is used in heterogeneous Cloud infrastructures to decrease the makespan of Cloud infrastructure and achieve an energy-efficient environment. R Chaudhary et al. [30] proposed an SDN-based BD management scenario in regard to optimized network resource consumption, such as network bandwidth and data storage units. Finally, Y. Wen et al. [31] proposed MOPA in regard to the problem of scheduling workflow under security constraints, along with simultaneously reducing the execution time and the cost for BD applications in a Cloud infrastructure.

Table 7 shows that most of the related works involved and tried to solve the challenges related to *Computation (Processing) and Analysis*. Furthermore, most of the former related works dealt with *Privacy*, *Security*, and *Management*. The challenges that were not a main research interest were *Energy Efficiency* and *Access Control*. Consequently, we can conclude that there are many open issues in the field of CC and BD integration that need to be solved. To sum up, there is a need for more research in the area of Security and Management of BD in a CC environment, and thus, we propose a new encryption method of the data in a sustainable Cloud environment that will deal with and solve challenges such as the Management and Security of data, which, in many cases, are IoT-based Big Data.

**Table 7.** Related work comparison.

| Big Data and Cloud Computing Integration Challenges | Privacy | Security | Energy Efficient | Access Control | Computation (Processing) and Analysis | Management |
|---|---|---|---|---|---|---|
| Gai et al. [28] | √ | √ | | | | |
| Mishra et al. [29] | | | √ | | √ | |

**Table 7.** *Cont.*

| Big Data and Cloud Computing Integration Challenges | Privacy | Security | Energy Efficient | Access Control | Computation (Processing) and Analysis | Management |
|:---:|:---:|:---:|:---:|:---:|:---:|:---:|
| Chaudhary et al. [30] | | | | √ | √ | √ |
| Wen et al. [31] | √ | √ | | | | |
| Proposed method | √ | √ | √ | √ | √ | √ |

## 7. Experimental Results and Analysis

To prove the beneficial operation of our proposed model, 10 simulations were conducted. With the experimental scenarios that we carried out, we compared the function of our proposed model with existing AES, RC5, and RSA algorithms. Through these simulations and their results, it can be concluded that our model is more secure and efficient.

Figures 4–7 show four experimental scenarios that considered the performance of data processing (in megabytes—MB) with the use of the four algorithms (AES, RC5, RSA, and the proposed model) in the measure of time. Each figure shows a different scenario of the amount of data used. Through these scenarios, we observed that by applying our proposed model, we could achieve better/faster data processing in comparison to the existing encryption algorithms AES, RC5, and RSA, as it could process a larger amount of data at the same time. In the figures, the legend is as follows: the red line represents AES, the green line represents RC5, the yellow line represents RSA, and the blue line represents our proposed model.

## 8. Conclusions

Here, we presented a study of the CC and IoT-based BD, aiming to tackle their current challenges in terms of security and management. Specifically, we combined them to determine their related characteristics and the benefits of their integration. Subsequently, we presented the contribution of Big Data to CC, aiming to fill the existing scientific gap in this field. Moreover, this work shows how CC improved the function of IoT-based BD. Finally, we investigated the security issues of CC and BD integration and proposed a novel security model for a more sustainable environment. The experimental results on the use of the encryption algorithms AES, RC5, RSA, and the proposed model, extending the advances of Cloud Computing and IoT-based Big Data, offer a highly novel and scalable efficient service platform to achieve more privacy and security services.

By providing an overview of the current challenges in the use of IoT-based Big Data in Cloud Computing while also presenting a Digital Twin scenario with a novel security model and an algorithm for sustainable Cloud systems, we have provided a solid framework for real-life applications that constitute the top priorities for industry and authorities alike. For instance, the EU has declared its goal to provide readily available ocean knowledge to various stakeholders through an innovative Digital Twin tool (European Digital Twin of the Ocean; *https://research-and-innovation.ec.europa.eu/funding/funding-opportunities/funding-programmes-and-open-calls/horizon-europe/eu-missions-horizon-europe/restore-our-ocean-and-waters/european-digital-twin-ocean-european-dto_en*(accessed on 27 December 2022)) to assist in the restoration of marine and coastal habitats, the sustainable blue economy, and the mitigation of climate change. Our work ensures that any digital representation of the past and present state of real-world entities and/or processes using real-time and historical data and creating models to simulate future scenarios will be based on enhanced security and a more streamlined data management protocol.

Finally, the security problems of Cloud Computing and IoT-based Big Data integration were studied through the proposed algorithm model. In addition to this, our work shows

how the three encryption algorithms analyzed here contribute to the integrated model of CC and IoT-based BD. Consequently, based on this, we can determine the scope of future research on the integration of CC and IoT-based BD. Due to their hasty development, the security and management challenges of data streamed and stored in a sustainable Cloud must be clarified or minimized to create an efficient integrated model. The security issues studied here could be an area for future research as a case study, with the intention of minimizing them.

**Author Contributions:** Validation, C.L.S., E.B. and K.E.P.; Formal analysis, C.L.S.; Resources, C.L.S.; Writing—original draft, C.L.S.; Writing—review & editing, C.L.S., E.B.; Visualization, C.L.S.; Supervision, K.E.P.; Project administration, C.L.S. All authors have read and agreed to the published version of the manuscript.

**Funding:** This research received no external funding.

**Institutional Review Board Statement:** Not applicable.

**Informed Consent Statement:** Not applicable.

**Data Availability Statement:** Not applicable.

**Conflicts of Interest:** The authors declare no conflict of interest.

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
