# Peer review of "Security and Privacy Issues in IoT-Based Big Data Cloud Systems in a Digital Twin Scenario"

_applsci, doi:10.3390/app13020758_

Round 1

Reviewer 1 Report

The manuscript by Stergiou and Psannis provides an overview of the current challenges in of Cloud Computing approaches, when treating with IoT-based Big Data. Furthermore, the authors present a Digital Twin Scenario consisting of a novel security model and an algorithm for sustainable Cloud systems, with the intention to offer a more secure use of Big Data in Cloud Computing. The arguments offered are clear and well-constructed, demonstrating the favorable metrics of their customized approach, which compares well to the established AES, RC5, and RSA encryption algorithms. The topic of secure and efficient data integration is of paramount importance for the evolution of the vast majority of modern scientific and industrial fields, and should be of high interest to the audience of Applied Sciences, expert and non-expert. In my opinion, it will especially be a timely article for the mid- and long-term strategies of ocean and outer space exploration and management, which constitute the current frontiers in the next expansion phase of human activities.

I enjoyed reading this manuscript, and I would recommend it for publication after addressing the minor points presented below:

General comments:

Coming from a marine research background myself, I consider the topic highly relevant to the new generation of applied sciences, with the massive amounts and high complexity of the data generated to cover the needs of initiatives such as the UN’s Decade of Ocean Science for Sustainable Development (2021-2030). Novel robotic technologies are being developed and are soon to be deployed in real-life ecological monitoring scenarios, such as restoration of Marine Protected Areas, surveillance of deep-sea mining sites, etc.

In this framework, it would be very interesting to review the current state of the art in Cloud Computing, IoT, Big Data and Digital Twins within the marine domain, in a follow-up work with more advanced statistical analysis (e.g., Term-Map analysis).

I also believe that a bit more focus should be placed on such potential real-life applications, maybe expanding with an extra paragraph or short section before the conclusions. In my opinion, the manuscript would benefit from highlighting these aspects, and appeal more to the general audience than in its current, technically-oriented form.

I would suggest a thorough language revision, in order to improve the fluidity of the reading experience and for consistency of the used terms.

Specific points:

L10: Should read “…applications, and their data…”.

L11-12: Replace “consisted” with “consists”.

L28: Replace “…could offer some…” with “…can offer several….

L29-30: Unnecessary repetition of the term “Cloud environment”, the sentence could read “In particular, Cloud providers and the customers are keen to build a more secure Cloud environment, due to its unique function”.

L33-34: The abbreviated form “CCF” should be previously defined at least once in the main text.

L37: The start of the sentence could read “These data, if processed properly, could reveal…

L41: Replace “to” with “for”.

L42: Replace “consist of” with “constitute”. Add “(BD)” after “Big Data”, so that the abbreviation is defined in the main text apart from the abstract.

L43: Delete “the current”.

L48: Replace “defined” with “characterized”, as the text that follows is not a definition.

L49: Delete “which is”.

L50 (and throughout the text): I would suggest avoiding the back and forth between the full term and the abbreviations, if you start using the abbreviations you can keep them throughout the text (or maybe use the full term the first time it’s mentioned in each section).

L49-52: The sentences could be merged to “All the data related to the term Big Data have a specific origin or “source” which, based on Variety, could give various data types”.

L52-54: The sentences could be merged to “Here, we attempt to address several major Big Data sources (BDS), as well as the respective challenges that arise from them regarding the overall use of Big Data”.

L55: I’d suggest upgrading this category (e.g., Earth and Space Sciences?), as well as including the marine domain. Ocean cabled observatories generate multiparametric numeric and audiovisual data which are transferred and stored in real time. A characteristic example would be the observatories operated by Ocean Networks Canada (www.oceannetworks.ca) with the Oceans 3.0 data portal. Same for underwater neutrino telescopes, such as KM3NeT in the Mediterranean (www.km3net.org/). These research infrastructures generate massive amounts of data and could benefit from advancements in data management applications based on Cloud computing to streamline data quality control and integration and to increase security.

Potential references to be included:

·       Chatzievangelou D., Bahamon N., Martini S., del Rio J., Riccobene G., Tangherlini M., et al. (2021) Integrating Diel Vertical Migrations of Bioluminescent Deep Scattering Layers Into Monitoring Programs. Front. Mar. Sci. 8:661809. doi: 10.3389/fmars.2021.661809

·       Aguzzi J., Flögel S., Marini S., Thomsen L., Albiez J., Weiss P., et al. (2022). Developing Technological Synergies Between Deep-sea and Space research. Elem. Sci. Anth. 10:00064. doi: 10.1525/elementa.2021.00064

L61: Replace “more wide” with “wider”.

L65: Replace “describe” with “be described”.

L66: Delete “and”.

L67: Replace “…lead us in…” with “…lead to…”.

L69: Replace “…could also be generated…” with “…are being generated…”.

L77: Replace “…which is a…” with “…which is considered a…”.

L82-83: The sentences could be merged to “…used a Digital Twin Scenario with the goal to simulate the operation of a Cloud server, through which it manages…”.

L103: Replace “Proposes” with “Propose”.

L106-107: Replace “…researches which have been made in the field…” with “…research in the field…”.

L108: Replace “gives” with “provides” and “background researches” with “background studies”.

L111: Replace “related researches” with “related studies”.

L114: The start of the sentence could read “Next, Section 6 compares our…”.

L120: Replace “for this research” with “for this study”.

L121: Replace “previous literature researches” with “previous research from literature”.

L124: The start of the sentence could read “First off, Takabi et al…”.

L133: Delete the comma after “as well as”.

Table 1: There’s a strange graphic error between the columns “Security” and “Storage”

L164-165: The start of the sentence could read “Most of the works that listed in Table 1 focus on the “Management” challenge, which…”.

L166-167: Delete “As a result”. “Computation (Processing) & Analysis”, “Security” and “Privacy” issues being second is not a result of “Management” being first.

L174: Replace “…and also count on the recent…” with “…and taking also into account the recent…”.

L188-190: Same as in the intro, the abbreviated form “CCC” should be previously defined at least once in the main text.

Table 4: There’s a strange graphic error between the columns “Security” and “Storage”.

L290-292: This sentence seems to be repeating the information of the phrase in L261-263, consider rephrasing or deleting it.

L358: “…many simulations have been made” how many?

L362: “…a good effort has been done…” maybe try “successful”? Also, replace “done” with “made”.

Figures 4 to 7: What are the units for Data processed? Also, you cannot have the same caption in four figures, there should be more detail included in order to distinguish the four different experimental scenarios.

L375: “…we could achieve better data processing…” Maybe try “faster”? I would skip the next sentence (see my next comment) and merge this one with the last of the paragraph, to get “Through these scenarios, we can observe that by applying our proposed model we achieved faster data processing in comparison to the existed encryption algorithms AES, RC5, and RSA, as it could process a larger amount of data at the same time”.

L376-378: These clarifications should be integrated in the figure captions.

L382-383: The first sentence could read “Here we presented a study of CC and IoT-based BD, aiming to tackle their current challenges in terms of security and management”.

L384: “Consequently…” maybe try “Subsequently…”?

L393-401: This paragraph needs some rephrasing, to make the take-home message of this work easier to understand.

Author Response

Dear reviewer,

We would like to thank you for your helpful comments on the paper. We appreciate the effort you put on, and the time you devoted to writing your suggestions. We believe, we have been able to address all your comments and correct any deficiencies you have pointed out.

  1. In this framework, it would be very interesting to review the current state of the art in Cloud Computing, IoT, Big Data and Digital Twins within the marine domain, in a follow-up work with more advanced statistical analysis (e.g., Term-Map analysis, see Aguzzi et al., 2021. https://doi.org/10.3390/s21113778 and references therein).

Thank you for this direction. We appreciate your comment and we discussed the work of Aguzzi et al [https://doi.org/10.3390/s21113778] in the Section 2: Background Research, as shown in the following paragraph.

[Section 2, Paragraph 12]

Moreover, Aguzzi et al [8], through their research, focused on marine biomimetic research and used innovative bibliographic statistics tools, in order to highlight established and emerging knowledge domains. In the research environment of their work, they identified natural processes by which living organisms obtain energy is thus urgent to sustain energy-demanding tasks while, at the same time, the natural designs must increasingly inform to optimize energy consumption.

  1. I also believe that a bit more focus should be placed on such potential real-life applications, maybe expanding with an extra paragraph or short section before the conclusions. In my opinion, the manuscript would benefit from highlighting these aspects, and appeal more to the general audience than in its current, technically-oriented form.

Thank you for this advice. We appreciate your comment and we provide a better description of such potential real-life applications. We have endeavored to extend the discussion of potential real-life applications of our study in the following paragraph.

[Section 8, Paragraphs 2]

By providing an overview of the current challenges in the use of IoT-based Big Data in Cloud Computing, while also presenting a Digital Twin Scenario with a novel security model and an algorithm for sustainable Cloud systems, we intend to provide a solid framework for real-life applications which constitute top priorities for industry and authorities alike. For instance, the EU has declared its ambition to provide readily available ocean knowledge to various stakeholders, through an innovative Digital Twin tool (European Digital Twin of the Ocean; https://research-and-innovation.ec.europa.eu/funding/funding-opportunities/funding-programmes-and-open-calls/horizon-europe/eu-missions-horizon-europe/restore-our-ocean-and-waters/european-digital-twin-ocean-european-dto_en), to assist the restoration of marine and coastal habitats, the sustainable blue economy and the mitigation of climate change. Our work ensures that any digital representation of the past and present state of real-world entities and/or processes using real-time and historical data and creating models to simulate future scenarios, will count on enhanced security and a more stream-lined data management protocol.

  1. I would suggest a thorough language revision, in order to improve the fluidity of the reading experience and for consistency of the used terms.

Thank you for this advice. We appreciate your comment and we provide a better text by correcting all the points you mentioned.

  1. Potential references to be included:
    • Chatzievangelou D., Bahamon N., Martini S., del Rio J., Riccobene G., Tangherlini M., et al. (2021) Integrating Diel Vertical Migrations of Bioluminescent Deep Scattering Layers Into Monitoring Programs. Front. Mar. Sci. 8:661809. doi: 10.3389/fmars.2021.661809
    • Aguzzi J., Flögel S., Marini S., Thomsen L., Albiez J., Weiss P., et al. (2022). Developing Technological Synergies Between Deep-sea and Space research. Elem. Sci. Anth. 10:00064. doi: 10.1525/elementa.2021.00064

Thank you for your comment. We additionally cited the following references you have suggested:

  1. D. Chatzievangelou, N. Bahamon, S. Martini, J. del Rio, G. Riccobene, M. Tangherlini, R. Danovaro, F. C. De Leo, B. Pirenne, J. Aguzzi, “Integrating Diel Vertical Migrations of Bioluminescent Deep Scattering Layers Into Monitoring Programs”, Frontiers in Marine Science, vol. 8, pp. 661809, May 2021. [DOI: 10.3389/fmars.2021.661809]
  2. J. Aguzzi, S. Flögel, S. Marini, L. Thomsen, J. Albiez, P. Weiss, G. Picardi, M. Calisti, S. Stefanni, L. Mirimin, F. Vecchi, C. Laschi, A. Branch, E. B. Clark, B. Foing, A. Wedler, D. Chatzievangelou, M. Tangherlini, A. Purser, L. Dartnell, R. Da-novaro, “Developing technological synergies between deep-sea and space research”, Elementa: Science of the Anthropocene, vol. 10, issue: 1, pp. 1-19, February 2022. [DOI: 10.1525/elementa.2021.00064]
  3. J. Aguzzi, C. Costa, M. Calisti, V. Funari, S. Stefanni, R. Danovaro, H. I. Gomes, F. Vecchi, L. R. Dartnell, P. Weiss, K. Nowak, D. Chatzievangelou, S. Marini, “Research Trends and Future Perspectives in Marine Biomimicking Robotics”, MDPI, Sensors, vol. 21, issue: 11, pp. 3778, May 2021. [DOI: 10.3390/s21113778]

  1. Table 1: There’s a strange graphic error between the columns “Security” and “Storage”.

Thank you for this advice. We appreciate your comment and we correct the faulty graphic on the table.

  1. Table 4: There’s a strange graphic error between the columns “Security” and “Storage”.

Thank you for this advice. We appreciate your comment and we correct the faulty graphic on the table.

  1. Figures 4 to 7: What are the units for Data processed? Also, you cannot have the same caption in four figures, there should be more detail included in order to distinguish the four different experimental scenarios.

Thank you for this advice. We appreciate your comment and we provide a better description of the Figures by adding the units of the data processed in each of them. Additionally, we clarify the units of each figure in the text bellow of them.

The whole corrections or additions we have made in the paper we noticed with blue color.

Thank you very much for your attention and kind consideration.

Sincerely yours,

Dr. Christos L. Stergiou

Ph.D. University of Macedonia

Dept. of Applied Informatics

University of Macedonia

Thessaloniki, Greece

Email: c.stergiou@uom.edu.gr

Guest Editor MDPI Applied Sciences Special Issue “Secure Integration of IoT & Digital Twins”

https://www.mdpi.com/journal/applsci/special_issues/1N3J910V3M

Guest Editor MDPI Applied Sciences Special Issue “Application of Data Analytics in Smart Healthcare”

https://www.mdpi.com/journal/applsci/special_issues/Application_Data_Analytics_Smart_Healthcare

Elisavet Bompoli

Dept. of Applied Informatics

University of Macedonia

Thessaloniki, Greece

Email: ebompoli@gmail.com

Prof. Kostas E. Psannis

Associate Editor IEEE Access

Associate Editor IEEE Com Letters

Guest Editor MDPI Applied Sciences Special Issue “Secure Integration of IoT & Digital Twins”

https://www.mdpi.com/journal/applsci/special_issues/1N3J910V3M

Guest Editor MDPI Applied Sciences Special Issue “Application of Data Analytics in Smart Healthcare”

https://www.mdpi.com/journal/applsci/special_issues/Application_Data_Analytics_Smart_Healthcare

Guest Editor MDPI Applied Sciences Special Issue “Advancements in QoS/QoE for Future Networks and Their Applications” https://www.mdpi.com/journal/applsci/special_issues/PPMUH0SF3O

Guest Editor MDPI Applied Sciences Special Issue “Application of Data Analytics in Smart Healthcare” https://www.mdpi.com/journal/applsci/special_issues/Application_Data_Analytics_Smart_Healthcare

Guest Editor MDPI Telecom Special Issue “Papers from the 4th World Symposium on Communication Engineering (WSCE 2021)” https://www.mdpi.com/journal/telecom/special_issues/SI_WSCE2021

Guest Editor MDPI Applied Sciences Topical Collection “5G Networks: Optimization, Machine Learning and Blockchain Technologies II” https://www.mdpi.com/journal/applsci/sections/computing_artificial_intelligence

Guest Editor MDPI Telecom Special Issue “6G Wireless Communication Systems” https://www.mdpi.com/journal/telecom/special_issues/6g_wireless_communication

Guest Editor MDPI Sensors Special Issue “Compressive Sensing-Based IoT Applications” https://www.mdpi.com/journal/sensors/special_issues/Compressive_Sensing_IoT

Guest Editor MDPI Information Special Issue “ICCCI 2020&2021: Advances in Baseband Signal Processing, Circuit Designs, and Communications” https://www.mdpi.com/journal/information/special_issues/ICCCI_2020

Guest Editor MDPI Applied Sciences Special Issue “5G Networks: Optimization, Machine Learning And Blockchain Technologies” https://www.mdpi.com/journal/applsci/special_issues/5G_blockchain

Guest Editor MDPI Sensors Special Issue “6G Wireless Communication Systems” https://www.mdpi.com/journal/sensors/special_issues/6g_wireless_communication_systems’

Guest Editor MDPI Telecom Special Issue “Papers from the 4th World Symposium on Communication Engineering (WSCE 2021)” https://www.mdpi.com/journal/telecom/special_issues/SI_WSCE2021

Dept. of Applied Informatics

University of Macedonia

Thessaloniki, Greece

Email: kpsannis@uom.edu.gr

Tel: +30 2310 891 737

Reviewer 2 Report

Based on the analysis of the security challenges of the integrated BD and CC models, this paper constructs a new security algorithm. The experimental results show that this algorithm has more advantages than the traditional encryption algorithm. The content of this paper has a feasible contribution to the security and privacy protection of digital twin scenes.

However, the following problems still exist:

1. The abbreviations "SaaS" and "IaaS" in line 247 do not indicate the original meaning. It is a threshold for readers who do not know the cloud computing field, so it is recommended to make a brief introduction.

2. The specific connotation of "CCF1" - "CCF5" in Table 2 and "BDS1" - "BDS5" in Table 3 is not clearly stated, and there is a problem of unclear reference. It is suggested to make a proper introduction first to reduce the threshold for readers to read.

3. The proposed algorithm should be further described in detail, and the process should be given with a specific description. It is hoped to be further clarified in Section 5.3.

Author Response

Dear reviewer,

We would like to thank you for your helpful comments on the paper. We appreciate the effort you put on, and the time you devoted to writing your suggestions. We believe, we have been able to address all your comments and correct any deficiencies you have pointed out.

  1. The abbreviations "SaaS" and "IaaS" in line 247 do not indicate the original meaning. It is a threshold for readers who do not know the cloud computing field, so it is recommended to make a brief introduction.

Thank you for this direction. We appreciate your comment and we provide a better description of all abbreviations of our paper before their use.

  1. The specific connotation of "CCF1" - "CCF5" in Table 2 and "BDS1" - "BDS5" in Table 3 is not clearly stated, and there is a problem of unclear reference. It is suggested to make a proper introduction first to reduce the threshold for readers to read.

Thank you for this direction. We appreciate your comment. We have tried to present better all abbreviations of our paper by proper introduction them.

  1. The proposed algorithm should be further described in detail, and the process should be given with a specific description. It is hoped to be further clarified in Section 5.3.

Thank you for this advice. We have endeavored to extend the discussion of the proposed algorithm of our study in the following paragraph. More specific, we noticed the computational complexity of it in the following paragraph.

[Subsection 5.3, Paragraph 4]

We decide to improve the existing benefits of the three aforementioned algorithms by proposing a novel one, which combines their major benefits of them as a new algorithm scenario. To prove the functionality of the proposed algorithm we have made several simulations with different amounts of data and through various time scenarios to study in a better way the complexity of our algorithm. Experimental scenarios and results are illustrated in Section 7, and through them shows how the proposed algorithm operates in a better way than the existing algorithms. We present four different scenarios (figures 4 to 7), showing four different combinations of data and time.

The whole corrections or additions we have made in the paper we noticed with blue color.

Thank you very much for your attention and kind consideration.

Sincerely yours,

Dr. Christos L. Stergiou

Ph.D. University of Macedonia

Dept. of Applied Informatics

University of Macedonia

Thessaloniki, Greece

Email: c.stergiou@uom.edu.gr

Guest Editor MDPI Applied Sciences Special Issue “Secure Integration of IoT & Digital Twins”

https://www.mdpi.com/journal/applsci/special_issues/1N3J910V3M

Guest Editor MDPI Applied Sciences Special Issue “Application of Data Analytics in Smart Healthcare”

https://www.mdpi.com/journal/applsci/special_issues/Application_Data_Analytics_Smart_Healthcare

Elisavet Bompoli

Dept. of Applied Informatics

University of Macedonia

Thessaloniki, Greece

Email: ebompoli@gmail.com

Prof. Kostas E. Psannis

Associate Editor IEEE Access

Associate Editor IEEE Com Letters

Guest Editor MDPI Applied Sciences Special Issue “Secure Integration of IoT & Digital Twins”

https://www.mdpi.com/journal/applsci/special_issues/1N3J910V3M

Guest Editor MDPI Applied Sciences Special Issue “Application of Data Analytics in Smart Healthcare”

https://www.mdpi.com/journal/applsci/special_issues/Application_Data_Analytics_Smart_Healthcare

Guest Editor MDPI Applied Sciences Special Issue “Advancements in QoS/QoE for Future Networks and Their Applications” https://www.mdpi.com/journal/applsci/special_issues/PPMUH0SF3O

Guest Editor MDPI Applied Sciences Special Issue “Application of Data Analytics in Smart Healthcare” https://www.mdpi.com/journal/applsci/special_issues/Application_Data_Analytics_Smart_Healthcare

Guest Editor MDPI Telecom Special Issue “Papers from the 4th World Symposium on Communication Engineering (WSCE 2021)” https://www.mdpi.com/journal/telecom/special_issues/SI_WSCE2021

Guest Editor MDPI Applied Sciences Topical Collection “5G Networks: Optimization, Machine Learning and Blockchain Technologies II” https://www.mdpi.com/journal/applsci/sections/computing_artificial_intelligence

Guest Editor MDPI Telecom Special Issue “6G Wireless Communication Systems” https://www.mdpi.com/journal/telecom/special_issues/6g_wireless_communication

Guest Editor MDPI Sensors Special Issue “Compressive Sensing-Based IoT Applications” https://www.mdpi.com/journal/sensors/special_issues/Compressive_Sensing_IoT

Guest Editor MDPI Information Special Issue “ICCCI 2020&2021: Advances in Baseband Signal Processing, Circuit Designs, and Communications” https://www.mdpi.com/journal/information/special_issues/ICCCI_2020

Guest Editor MDPI Applied Sciences Special Issue “5G Networks: Optimization, Machine Learning And Blockchain Technologies” https://www.mdpi.com/journal/applsci/special_issues/5G_blockchain

Guest Editor MDPI Sensors Special Issue “6G Wireless Communication Systems” https://www.mdpi.com/journal/sensors/special_issues/6g_wireless_communication_systems’

Guest Editor MDPI Telecom Special Issue “Papers from the 4th World Symposium on Communication Engineering (WSCE 2021)” https://www.mdpi.com/journal/telecom/special_issues/SI_WSCE2021

Dept. of Applied Informatics

University of Macedonia

Thessaloniki, Greece

Email: kpsannis@uom.edu.gr

Tel: +30 2310 891 737
